# DISCRIMINATOR REJECTION SAMPLING

**Samaneh Azadi**[*]
UC Berkeley

**Catherine Olsson**
Google Brain

**Trevor Darrell**
UC Berkeley

**Ian Goodfellow**
Google Brain

**Augustus Odena**
Google Brain

## ABSTRACT

We propose a rejection sampling scheme using the discriminator of a GAN to approximately correct errors in the GAN generator distribution. We show that under quite strict assumptions, this will allow us to recover the data distribution exactly. We then examine where those strict assumptions break down and design a practical algorithm—called Discriminator Rejection Sampling (DRS)—that can be used on real data-sets. Finally, we demonstrate the efficacy of DRS on a mixture of Gaussians and on the state of the art SAGAN model. On ImageNet, we train an improved baseline that increases the best published Inception Score from 52.52 to 62.36 and reduces the Fréchet Inception Distance from 18.65 to 14.79. We then use DRS to further improve on this baseline, improving the Inception Score to 76.08 and the FID to 13.75.

## 1 INTRODUCTION

Generative Adversarial Networks (GANs) (Goodfellow et al., 2014) are a powerful tool for image synthesis. They have also been applied successfully to semi-supervised and unsupervised learning (Springenberg, 2015; Odena, 2016; Kumar et al., 2017), image editing (Yu et al., 2018; Ledig et al., 2017), and image style transfer (Zhu et al., 2017; Isola et al., 2017; Yi et al., 2017; Azadi et al., 2018). Informally, the GAN training procedure pits two neural networks against each other, a generator and a discriminator. The discriminator is trained to distinguish between samples from the target distribution and samples from the generator. The generator is trained to fool the discriminator into thinking its outputs are real. The GAN training procedure is thus a two-player differentiable game, and the game dynamics are largely what distinguishes the study of GANs from the study of other generative models. These game dynamics have well-known and heavily studied stability issues. Addressing these issues is an active area of research (Mao et al., 2017; Arjovsky et al., 2017; Gulrajani et al., 2017; Odena et al., 2018; Li et al., 2017).

However, we are interested in studying something different: Instead of trying to improve the training procedure, we (temporarily) accept its flaws and attempt to improve the quality of trained generators by post-processing their samples using information from the trained discriminator. It's well known that (under certain very strict assumptions) the equilibrium of this training procedure is reached when sampling from the generator is identical to sampling from the target distribution and the discriminator always outputs $1/2$. However, these assumptions don't hold in practice. In particular, GANs as presently trained don't learn to reproduce the target distribution (Arora & Zhang, 2017). Moreover, trained GAN discriminators aren't just identically $1/2$ — they can even be used to perform chess-type skill ratings of other trained generators (Olsson et al., 2018).

We ask if the information retained in the weights of the discriminator at the end of the training procedure can be used to "improve" the generator. At face value, this might seem unlikely. After all, if there is useful information left in the discriminator, why doesn't it find its way into the generator via the training procedure? Further reflection reveals that there are many possible reasons. First, the assumptions made in various analyses of the training procedure surely don't hold in practice (e.g. the discriminator and generator have finite capacity and are optimized in parameter space rather than density-space). Second, due to the concrete realization of the discriminator and the generator as

---

[*]Correspondence to `sazadi@cs.berkeley.edu`

neural networks, it may be that it is harder for the generator to model a given distribution than it is for the discriminator to tell that this distribution is not being modeled precisely. Finally, we may simply not train GANs long enough in practice for computational reasons.

In this paper, we focus on using the discriminator as part of a probabilistic rejection sampling scheme. In particular, this paper makes the following contributions:

- We propose a rejection sampling scheme using the GAN discriminator to approximately correct errors in the GAN generator distribution.

- We show that under quite strict assumptions, this scheme allows us to recover the data distribution exactly.

- We then examine where those strict assumptions break down and design a practical algorithm – called DRS – that takes this into account.

- We conduct experiments demonstrating the effectiveness of DRS. First, as a baseline, we train an improved version of the Self-Attention GAN, improving its performance from the best published Inception Score of 52.52 up to 62.36, and from a Fréchet Inception Distance of 18.65 down to 14.79. We then show that DRS yields further improvement over this baseline, increasing the Inception Score to 76.08 and decreasing the Fréchet Inception Distance to 13.75.

## 2 BACKGROUND

### 2.1 GENERATIVE ADVERSARIAL NETWORKS

A generative adversarial network (GAN) (Goodfellow et al., 2014) consists of two separate neural networks — a generator, and a discriminator — trained in tandem. The generator $G$ takes as input a sample from the prior $z \in Z \sim p_z$ and produces a sample $G(z) \in X$. The discriminator takes an observation $x \in X$ as input and produces a probability $D(x)$ that the observation is real. The observation is sampled either according to the density $p_d$ (the data generating distribution) or $p_g$ (the implicit density given by the generator and the prior). Using the standard non-saturating variant, the discriminator and generator are then trained using the following loss functions:

$$
\begin{aligned}
L_D &= -\mathbb{E}_{x \sim p_{\text{data}}}[\log D(x)] - \mathbb{E}_{z \sim p_z}[1 - \log D(G(z))] \\
L_G &= -\mathbb{E}_{z \sim p_z}[\log D(G(z))]
\end{aligned}
\tag{1}
$$

### 2.2 EVALUATION METRICS: INCEPTION SCORE (IS) AND FRÉCHET INCEPTION DISTANCE (FID)

The two most popular techniques for evaluating GANs on image synthesis tasks are the Inception Score and the Fréchet Inception Distance. The Inception Score (Salimans et al., 2016) is given by $\exp(\mathbb{E}_x \text{KL}(p(y|x)||p(y)))$, where $p(y|x)$ is the output of a pre-trained Inception classifier (Szegedy et al., 2014). This measures the ability of the GAN to generate samples that the pre-trained classifier confidently assigns to a particular class, and also the ability of the GAN to generate samples from all classes. The Fréchet Inception Distance (FID) (Heusel et al., 2017), is computed by passing samples through an Inception network to yield "semantic embeddings", after which the Fréchet distance is computed between Gaussians with moments given by these embeddings.

### 2.3 SELF-ATTENTION GAN

We use a Self-Attention GAN (SAGAN) (Zhang et al., 2018) in our experiments. We do so because SAGAN is considered state of the art on the ImageNet conditional-image-synthesis task (in which images are synthesized conditioned on class identity). SAGAN differs from a vanilla GAN in the following ways: First, it uses large residual networks (He et al., 2016) instead of normal convolutional layers. Second, it uses spectral normalization (Miyato et al., 2018) in the generator and the discriminator and a much lower learning rate for the generator than is conventional (Heusel et al., 2017). Third, SAGAN makes use of self-attention layers (Wang et al.), in order to better model

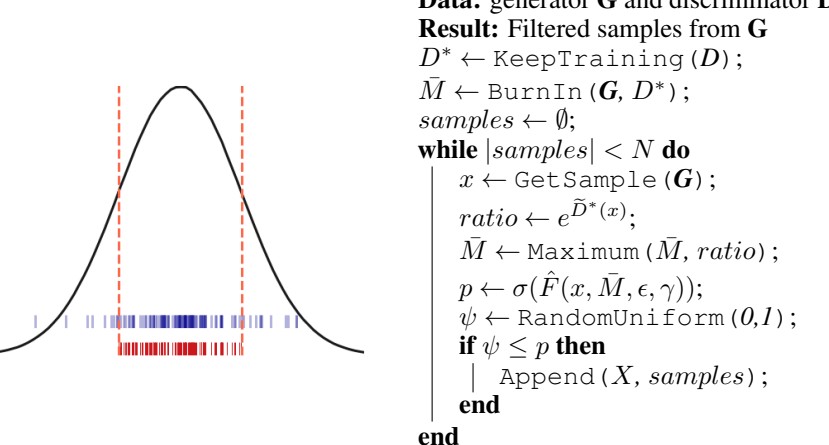

**Data:** generator **G** and discriminator **D**
**Result:** Filtered samples from **G**
$D^* \leftarrow$ `KeepTraining(D)`;
$\bar{M} \leftarrow$ `BurnIn(G, D*)`;
$samples \leftarrow \emptyset$;
**while** $|samples| < N$ **do**
    $x \leftarrow$ `GetSample(G)`;
    $ratio \leftarrow e^{\widetilde{D}^*(x)}$;
    $\bar{M} \leftarrow$ `Maximum(M̄, ratio)`;
    $p \leftarrow \sigma(\hat{F}(x, \bar{M}, \epsilon, \gamma))$;
    $\psi \leftarrow$ `RandomUniform(0,1)`;
    **if** $\psi \le p$ **then**
        `Append(X, samples)`;
    **end**
**end**

Figure 1: **Left:** For a uniform proposal distribution and Gaussian target distribution, the blue points are the result of rejection sampling and the red points are the result of naively throwing out samples for which the density ratio $(p_d(x)/p_g(x))$ is below a threshold. The naive method underrepresents the density of the tails. **Right:** the DRS algorithm. *KeepTraining* continues training using early stopping on the validation set. *BurnIn* computes a large number of density ratios to estimate their maximum. $\widetilde{D}^*$ is the logit of $D^*$. $\hat{F}$ is as in Equation 8. $\bar{M}$ is an empirical estimate of the true maximum $M$.

long range dependencies in natural images. Finally, this whole model is trained using a special hinge version of the adversarial loss (Lim & Ye, 2017; Miyato & Koyama, 2018; Tran et al., 2017):

$$L_D = -\mathbb{E}_{(x,y)\sim p_{\text{data}}}[\min(0, -1 + D(x,y))] - \mathbb{E}_{z\sim p_z, y\sim p_{\text{data}}}[\min(0, -1 - D(G(z),y))]$$
$$L_G = -\mathbb{E}_{z\sim p_z, y\sim p_{\text{data}}}[D(G(z),y))] \tag{2}$$

### 2.4 REJECTION SAMPLING

Rejection sampling is a method for sampling from a target distribution $p_d(x)$ which may be hard to sample from directly. Samples are instead drawn from a proposal distribution $p_g(x)$, which is easier to sample from, and which is chosen such that there exists a finite value $M$ such that $Mp_g(x) > p_d(x)$ for $\forall x \in \text{domain}(p_d(x))$. A given sample $y$ drawn from $p_g$ is kept with acceptance probability $p_d(y)/Mp_g(y)$, and rejected otherwise. See the blue points in Figure 1 (Left) for a visualization. Ideally, $p_g(x)$ should be close to $p_d(x)$, otherwise many samples will be rejected, reducing the efficiency of the algorithm (MacKay, 2003).

In Section 3, we explain how to apply this rejection sampling algorithm to the GAN framework: in brief, we draw samples from the trained generator, $p_g(x)$, and then reject some of those samples using the discriminator to attain a closer approximation to the true data distribution, $p_d(x)$. An independent rejection sampling approach was proposed by Grover et al. (2018) in the latent space of variational autoencoders for improving samples from the variational posterior.

## 3 REJECTION SAMPLING FOR GANS

In this section we introduce our proposed rejection sampling scheme for GANs (which we call Discriminator Rejection Sampling, or DRS). We'll first derive an idealized version of the algorithm that will rely on assumptions that don't necessarily hold in realistic settings. We'll then discuss the various ways in which these assumptions might break down. Finally, we'll describe the modifications we made to the idealized version in order to overcome these challenges.

### 3.1 REJECTION SAMPLING FOR GANS: THE IDEALIZED VERSION

Suppose that we have a GAN and our generator has been trained to the point that $p_g$ and $p_d$ have the same support. That is, for all $x \in X$, $p_g(x) \ne 0$ if and only if $p_d(x) \ne 0$. If desired, we can make $p_d$

and $p_g$ have support everywhere in $X$ if we add low-variance Gaussian noise to the observations. Now further suppose that we have some way to compute $p_d(x)/p_g(x)$. Then, if $M = \max_x p_d(x)/p_g(x)$, then $M p_g(x) > p_d(x)$ for all $x$, so we can perform rejection sampling with $p_g$ as the proposal distribution and $p_d$ as the target distribution as long as we can evaluate the quantity $p_d(x)/M p_g(x)$[1]. In this case, we can exactly sample from $p_d$ (Casella et al., 2004), though we may have to reject many samples to do so.

But how can we evaluate $p_d(x)/M p_g(x)$? $p_g$ is defined only implicitly. One thing we can do is to borrow an analysis from the original GAN paper (Goodfellow et al., 2014), which assumes that we can optimize the discriminator in the space of density functions rather than via changing its parameters. If we make this assumption, as well as the assumption that the discriminator is defined by a sigmoid applied to some function of $x$ and trained with a cross-entropy loss, then by Proposition 1 of that paper, we have that, for any fixed generator and in particular for the generator $G$ that we have when we stop training, training the discriminator to completely minimize its own loss yields

$$D^*(x) = \frac{p_d(x)}{p_d(x) + p_g(x)} \tag{3}$$

We will discuss the validity of these assumptions later, but for now consider that this allows us to solve for $p_d(x)/p_g(x)$ as follows: As noted above, we can assume the discriminator is defined as:

$$D(x) = \sigma(x) = \frac{1}{1 + e^{-\widetilde{D}(x)}}, \tag{4}$$

where $D(x)$ is the final discriminator output after the sigmoid, and $\widetilde{D}(x)$ is the logit. Thus,

$$
\begin{aligned}
D^*(x) = \frac{1}{1 + e^{-\widetilde{D}^*(x)}} &= \frac{p_d(x)}{p_d(x) + p_g(x)} \\
1 + e^{-\widetilde{D}^*(x)} &= \frac{p_d(x) + p_g(x)}{p_d(x)} \\
p_d(x) + p_d(x) e^{-\widetilde{D}^*(x)} &= p_d(x) + p_g(x) \\
p_d(x) e^{-\widetilde{D}^*(x)} &= p_g(x) \\
\frac{p_d(x)}{p_g(x)} &= e^{\widetilde{D}^*(x)}
\end{aligned}
\tag{5}
$$

Now suppose one last thing, which is that we can tractably compute $M = \max_x p_d(x)/p_g(x)$. We would find that $M = p_d(x^*)/p_g(x^*) = e^{\widetilde{D}^*(x^*)}$ for some (not necessarily unique) $x^*$. Given all these assumptions, we can now perform rejection sampling as promised. If we define $\widetilde{D}^*_M := \widetilde{D}^*(x^*)$, then for any input $x$, the acceptance probability $p_d(x)/M p_g(x)$ can be written as $e^{\widetilde{D}^*(x) - \widetilde{D}^*_M} \in [0, 1]$. To decide whether to keep any particular example, we can just draw a random number $\psi$ uniformly from $[0, 1]$ and accept the sample if $\psi < e^{\widetilde{D}^*(x) - \widetilde{D}^*_M}$.

## 3.2 DISCRIMINATOR REJECTION SAMPLING: THE PRACTICAL SCHEME

As we hinted at, the above analysis has a number of practical issues. In particular:

1. Since we can't actually perform optimization over density functions, we can't actually compute $D^*$. Thus, our acceptance probability won't necessarily be proportional to $p_d(x)/p_g(x)$.

2. At least on large datasets, it's quite obvious that the supports of $p_g$ and $p_d$ are not the same. If the support of $p_g$ and $p_d$ has a low volume intersection, we may not even want to compute $D^*$, because then $p_d(x)/p_g(x)$ would just evaluate to 0 most places.

3. The analysis yielding the formula for $D^*$ also assumes that we can draw infinite samples from $p_d$, which is not true in practice. If we actually optimized $D$ all the way given a finite data-set, it would give nonzero results on a set of measure 0.

---

[1] Why go through all this trouble when we could instead just pick some threshold $T$ and throw out $x$ when $D^*(x) < T$? This doesn't allow us to recover $p_d$ in general. If, for example, there is $x'$ s.t. $p_g(x') > p_d(x') > 0$, we still want some probability of observing $x'$. See the red points in Figure 1 (Left) for a visual explanation.

4. In general it won't be tractable to compute $M$.

5. Rejection sampling is known to have too low an acceptance probability when the target distribution is high dimensional (MacKay, 2003).

This section describes the Discriminator Rejection Sampling (DRS) procedure, which is an adjustment of the idealized procedure, meant to address the above issues.

**On the difficulty of actually computing $D^*$:**  Given that items 2 and 3 suggest we may not want to compute $D^*$ exactly, we should perhaps not be too concerned with item 1, which suggests that we can't. The best argument we can make that it is OK to approximate $D^*$ is that doing so seems to be successful empirically. We speculate that training a regularized $D$ with SGD gives a final result that is further from $D^*$ but perhaps is less over-fit to the finite sample from $p_d$ used for training. We also hypothesize that the $D$ we end up with will distinguish between "good" and "bad" samples, even if those samples would both have zero density under the true $p_d$. We qualitatively evaluate this hypothesis in Figures 4 and 5. We suspect that more could be done theoretically to quantify the effect of this approximation, but we leave this to future work.

**On the difficulty of actually computing $M$:**  It's nontrivial to compute $M$, at the very least because we can't compute $D^*$. In practice, we get around this issue by estimating $M$ from samples. We first run an estimation phase, in which 10,000 samples are used to estimate $\widetilde{D}^*_M$. We then use this estimate in the sampling phase. Throughout the sampling phase we update our estimate of $\widetilde{D}^*_M$ if a larger value is found. It's true that this will result in slight overestimates of the acceptance probability for samples that were processed before a new maximum was found, but we choose not to worry about this too much, since we don't find that we have to increase the maximum very often in the sampling phase, and the increase is very small when it does happen.

**Dealing with acceptance probabilities that are too low:**  Item 5 suggests that we may end up with acceptance probabilities that are too low to be useful when performing this technique on realistic data-sets. If $\widetilde{D}^*_M$ is very large, the acceptance probability $e^{\widetilde{D}^*(x)-\widetilde{D}^*_M}$ will be close to zero, and almost all samples will be rejected, which is undesirable. One simple way to avoid this problem is to compute some $F(x)$ such that the acceptance probability can be written as follows:

$$\frac{1}{1 + e^{-F(x)}} = e^{\widetilde{D}^*(x)-\widetilde{D}^*_M} \tag{6}$$

If we solve for $F(x)$ in the above equation we can then perform the following rearrangement:

$$
\begin{aligned}
F(x) &= \widetilde{D}^*(x) - \log(e^{\widetilde{D}^*_M} - e^{\widetilde{D}^*(x)}) \\
&= \widetilde{D}^*(x) - \log(\frac{e^{\widetilde{D}^*_M}}{e^{\widetilde{D}^*_M}}e^{\widetilde{D}^*_M} - \frac{e^{\widetilde{D}^*_M}}{e^{\widetilde{D}^*_M}}e^{\widetilde{D}^*(x)}) \\
&= \widetilde{D}^*(x) - \widetilde{D}^*_M - \log(1 - e^{\widetilde{D}^*(x)-\widetilde{D}^*_M})
\end{aligned}
\tag{7}
$$

In practice, we instead compute

$$\hat{F}(x) = \widetilde{D}^*(x) - \widetilde{D}^*_M - \log(1 - e^{\widetilde{D}^*(x)-\widetilde{D}^*_M-\epsilon}) - \gamma \tag{8}$$

where $\epsilon$ is a small constant added for numerical stability and $\gamma$ is a hyperparameter modulating overall acceptance probability. For very positive $\gamma$, all samples will be rejected. For very negative $\gamma$, all samples will be accepted. See Figure 2 for an analysis of the effect of adding $\gamma$. A summary of our proposed algorithm is presented in Figure 1 (Right).

## 4    EXPERIMENTS

In this section we justify the modifications made to the idealized algorithm. We do this by conducting two experiments in which we show that (according to popular measures of how well a GAN has

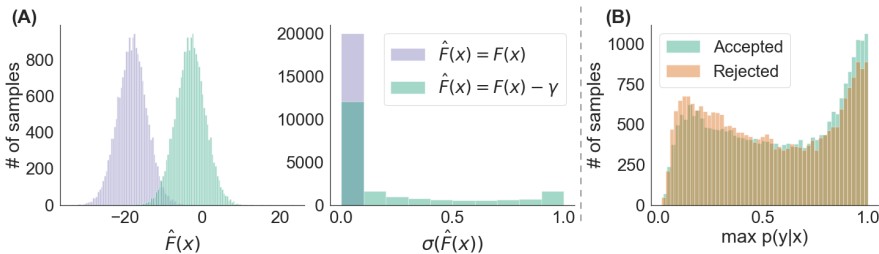

Figure 2: **(A)** Histogram of the sigmoid inputs, $\hat{F}(x)$ (left plot), and acceptance probabilities, $\sigma(\hat{F}(x))$ (center plot), on 20K fake samples before (purple) and after (green) adding the constant $\gamma$ to all $F(x)$. Before adding gamma, 98.9% of the samples had an acceptance probability $< 1e-4$. **(B)** Histogram of $\max_j p(y_j|x_i)$ from a pre-trained Inception network where $p(y_j|x_i)$ is the predicted probability of sample $x_i$ belonging to the $y_j$ category (from $1,000$ ImageNet categories). The green bars correspond to $25,000$ accepted samples and the red bars correspond to $25,000$ rejected samples. The rejected images are less recognizable as belonging to a distinct class.

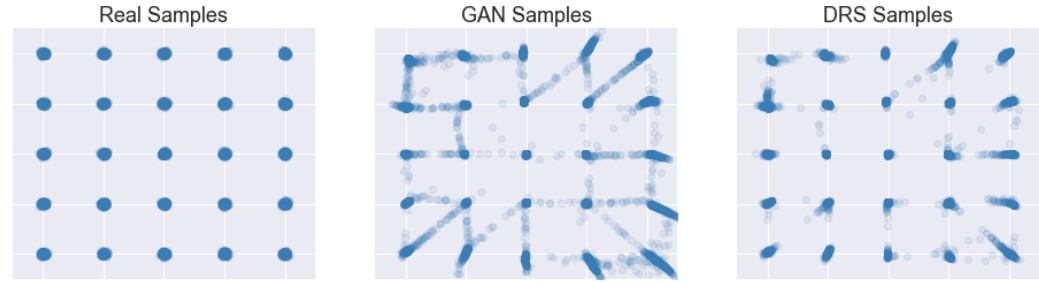

Figure 3: Real samples from 25 2D-Gaussian Distributions (*left*) as well as fake samples generated from a trained GAN model without (*middle*) and with DRS (*right*). Results are computed as an average over five models randomly initialized and trained independently.

learned the target distribution) Discriminator Rejection Sampling yields improvements for actual GANs. We start with a toy example that yields insight into how DRS can help, after which we demonstrate DRS on the ImageNet dataset (Russakovsky et al., 2015).

## 4.1 MIXTURE OF 25 GAUSSIANS

We investigate the impact of DRS on a low-dimensional synthetic data set consisting of a mixture of twenty-five 2D isotropic Gaussian distributions (each with standard deviation of 0.05) arranged in a grid (Dumoulin et al., 2016; Srivastava et al., 2017; Lin et al., 2017). We train a GAN model where the generator and discriminator are neural networks with four fully connected layers with ReLu activations. The prior is a 2D Gaussian with mean of 0 and standard deviation of 1 and the GAN is trained using the standard loss function. We generate 10,000 samples from the generator with and without DRS. The target distribution and both sets of generated samples are depicted in Figure 3. Here, we have set $\gamma$ dynamically for each batch, to the $95^{\text{th}}$ percentile of $\hat{F}(x)$ for all $x$ in the batch.

To measure performance, we assign each generated sample to its closest mixture component. As in Srivastava et al. (2017), we define a sample as "high quality" if it is within four standard deviations of its assigned mixture component. As shown in Table 1, DRS increases the fraction of high-quality samples from $70\%$ to $90\%$. As in Dumoulin et al. (2016) and Srivastava et al. (2017) we call a mode "recovered" if at least one high-quality sample was assigned to it. Table 1 shows that DRS does not reduce the number of recovered modes – that is, it does not trade off quality for mode coverage. It does reduce the standard deviation of the high-quality samples slightly, but this is a good thing in this case (since the standard deviation of the target Gaussian distribution is 0.05). It also confirms that DRS does not accept samples only near the center of each Gaussian but near the tails as well.

Table 1: Results with and without DRS on 10,000 generated samples from a model of a 2D grid of Gaussian components.

|  | # of recovered modes | % "high quality" | std of "high quality" samples |
|---|---|---|---|
| Without DRS | $24.8 \pm 0.4$ | $70 \pm 9$ | $0.11 \pm 0.01$ |
| With DRS | $24.8 \pm 0.4$ | $\mathbf{90 \pm 2}$ | $\mathbf{0.10 \pm 0.01}$ |

Table 2: Results with and without DRS on 50K ImageNet samples. Low FID and high IS are better.

|  | SAGAN | | Improved-SAGAN | |
|---|---|---|---|---|
|  | IS | FID | IS | FID |
| Without DRS | $52.34 \pm 0.45$ | $18.21 \pm 0.14$ | $62.36 \pm 0.35$ | $14.79 \pm 0.06$ |
| With DRS | $61.44 \pm 0.09$ | $17.14 \pm 0.09$ | $\mathbf{76.08 \pm 0.30}$ | $\mathbf{13.57 \pm 0.13}$ |

## 4.2 IMAGENET DATASET

Since it is presently the state-of-the-art model on the conditional ImageNet synthesis task, we have reimplemented the Self-Attention GAN (Zhang et al., 2018) as a baseline. After reproducing the results reported by Zhang et al. (2018) (with the learning rate of $1e^{-4}$), we fine-tuned a trained SAGAN with a much lower learning rate ($1e^{-7}$) for both generator and discriminator. This improved both the Inception Score and FID significantly as can be seen in the Improved-SAGAN column in Table 2. Plots of Inception score and FID during training are given in Figure 5(A).

Since SAGAN uses a hinge loss and DRS requires a sigmoid output, we added a fully-connected layer "on top of" the trained discriminator and trained it to distinguish real images from fake ones using the binary cross-entropy loss. We trained this extra layer with 10,000 generated samples from the model and 10,000 examples from ImageNet.

We then generated 50,000 samples from normal SAGAN and Improved SAGAN with and without DRS, repeating the sampling process 4 times. We set $\gamma$ dynamically to the $80^{\text{th}}$ percentile of the $F(x)$ values in each batch. The averages of Inception Score and FID over these four trials are presented in Table 2. Both scores were substantially improved for both models, indicating that DRS can indeed be useful in realistic settings involving large data-sets and sophisticated GAN variants.

**Qualitative Analysis of ImageNet results:** From a pool of 50,000 samples, we visualize the "best" and the "worst" 100 samples based on their acceptance probabilities. Figure 4 shows that the subjective visual quality of samples with high acceptance probability is considerably better. Figure 2(B) also shows that the accepted images are on average more recognizable as belonging to a distinct class.

We also study the behavior of the discriminator in another way. We choose an ImageNet category randomly, then generate samples from that category until we have found two images $G(z_1), G(z_2)$ such that $G(z_1)$ appears visually realistic and $G(z_2)$ appears visually unrealistic. Here, $z_1$ and $z_2$ are the input latent vectors. We then generate many images by interpolating in latent space between the two images according to $z = \alpha z_1 + (1 - \alpha)z_2$ with $\alpha \in \{0, 0.1, 0.2, \ldots, 1\}$. In Figure 5, the first and last columns correspond with $\alpha = 1$ and $\alpha = 0$, respectively. The color bar in the figure represents the acceptance probability assigned to each sample. In general, acceptance probabilities decrease from left to right. There is no reason to expect a priori that the acceptance probability should decrease monotonically as a function of the interpolated $z$, so it says something interesting about the discriminator that most rows basically follow this pattern.

## 5 CONCLUSION

We have proposed a rejection sampling scheme using the GAN discriminator to approximately correct errors in the GAN generator distribution. We've shown that under strict assumptions, we can recover the data distribution exactly. We've also examined where those assumptions break down and

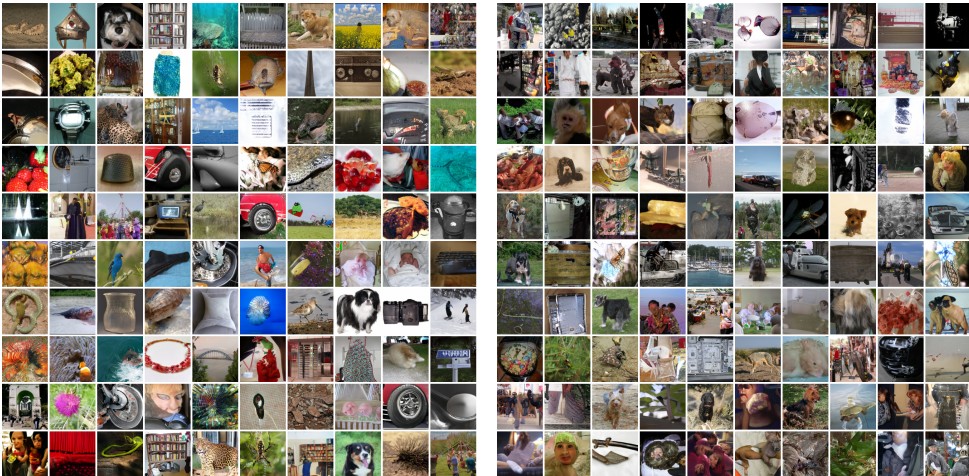

Figure 4: Synthesized images with the highest (*left*) and lowest (*right*) acceptance probability scores.

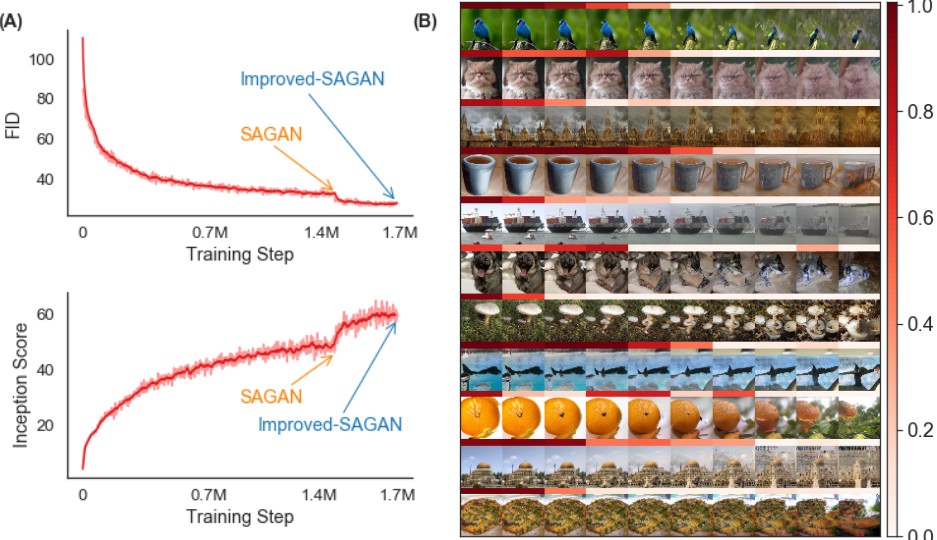

Figure 5: **(A)** Inception Score and FID during ImageNet training, computed on 50,000 samples. **(B)** Each row shows images synthesized by interpolating in latent space. The color bar above each row represents the acceptance probabilities for each sample: red for high and white for low. Subjective visual quality of samples with high acceptance probability is considerably better: objects are more coherent and more recognizable as belonging to a specific class. There are fewer indistinct textures, and fewer scenes without recognizable objects.

designed a practical algorithm (Discriminator Rejection Sampling) to address that. Finally, we have demonstrated the efficacy of this algorithm on a mixture of Gaussians and on the state-of-the-art SAGAN model.

Opportunities for future work include the following:

- There's no reason that our scheme can only be applied to GAN generators. It seems worth investigating whether rejection sampling can improve e.g. VAE decoders. This seems like it might help, because VAEs may have trouble with "spreading mass around" too much.
- In one ideal case, the critic used for rejection sampling would be a human. Can we use better proxies for the human visual system to improve rejection sampling's effect on image synthesis models?

- It would be interesting to theoretically characterize the efficacy of rejection sampling under the breakdown-of-assumptions that we have described earlier. For instance, if one can't recover $D^*$ but can train some other critic that has bounded divergence from $D^*$, how does the efficacy depend on this bound?

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

APPENDIX

## A  ABLATION STUDY

We have evaluated four different rejection sampling schemes on the mixture-of-Gaussians dataset, represented in Figure 6:

1. Always reject samples falling below a hard threshold and DO NOT train the Discriminator to "convergence".

2. Always reject samples falling below a hard threshold and train the Discriminator to convergence.

3. Use probabilistic sampling as in eq 8 and DO NOT train the Discriminator to convergence.

4. Our original DRS algorithm, in which we use probabilistic sampling and train the Discriminator to convergence.

In (1) and (2), we were careful to set the hard threshold so that the actual acceptance rate was the same as in (3) and (4). Broadly speaking, (4) performs best, (3) performs OK but yields less good samples than (4), (2) yields the same number of good samples as (3), but completely fails to sample from 5 of the 25 modes. (1) actually yields the most good samples for the modes it hits, but it only hits 4 modes!

These results show that both continuing to train $D$ so that it can approximate $D^*$ and performing sampling as in (8), which we have already motivated theoretically, is helpful in practice. For each method, we provide the number of samples within 1, 2, 3 and 4 standard deviations and the number of modes hit in Table 3. For reference, we also compute these statistics for the ground truth distribution and the unfiltered samples from GAN.

Table 3: Ablation study on 10,000 generated samples from a 2D grid of Gaussian components. The third to sixth columns represent % of high-quality samples within $x$ standard deviations. "No FT" stands for the discriminator not being trained to convergence.

|  | # of recovered modes | % in "1 std" | % in "2 std" | % in "3 std" | % in "4 std" |
|---|---|---|---|---|---|
| Ground Truth | 25 | 39.3 | 86.6 | 98.9 | 99.9 |
| Vanilla GAN | 25 | 27.3 | 53.1 | 66.2 | 75.6 |
| Threshold (No FT) | 4 | 38.5 | 92.6 | 99.4 | 99.8 |
| Threshold | 20 | 34.8 | 70.2 | 83.6 | 89.3 |
| DRS (No FT) | 25 | 31.5 | 60.2 | 73.6 | 81.2 |
| DRS | 25 | 35.3 | 65.8 | 81.8 | 89.8 |

In addition, we represent Inception score as a function of acceptance rate in Figure 7-left. Different acceptance rates are achieved by changing $\gamma$ from the $0^{th}$ percentile of $F(x)$ (acceptance rate = 100%) to its $90^{th}$ percentile (acceptance rate = 14%). Decreasing the acceptance rate filters more non-realistic samples and increases the final Inception score. After an specific rate, rejecting more samples does not gain any benefit in collecting a better pool of samples.

Moreover, Figure 7-right shows the correlation between the acceptance probabilities that DRS assigns to the synthesized samples and the recognizability of those samples from the view-point of a pre-trained Inception network. The latter is measured by computing $\max_j p(y_j|x_i)$ which is the probability of sample $x_i$ belonging to the category $y_j$ from the 1,000 ImageNet classes. As expected, there is a large mass of the recognizable images accepted with high acceptance probabilities on the top right corner. The small mass of images which cannot be easily classified into one of the 1,000 categories while having high acceptance probability scores (the top left corner of the graph) can be due to the non-optimal GAN discriminator in practice. Therefore, we expect that improving the discriminator performance boosts the final inception score even more substantially.

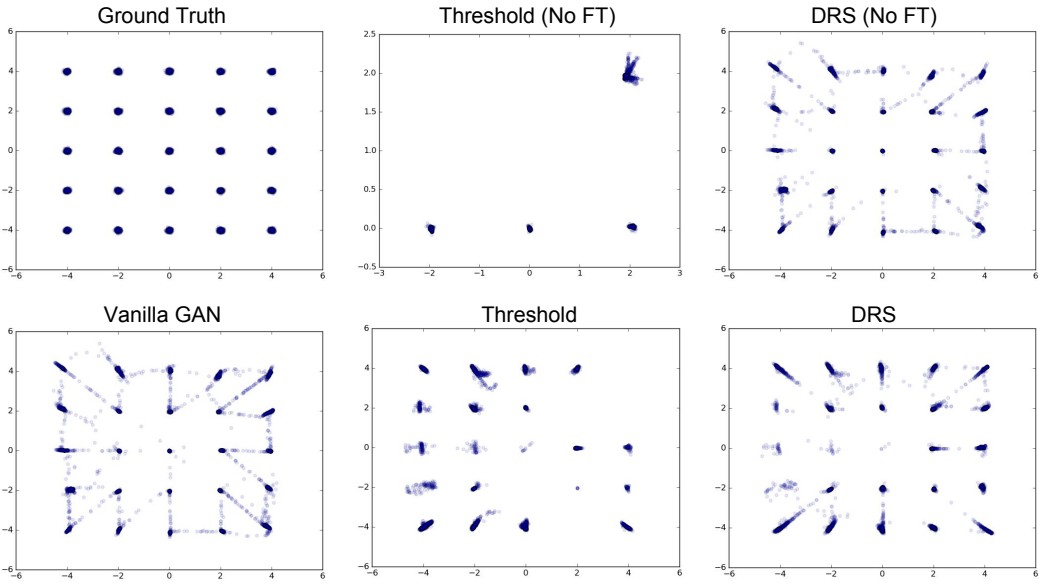

Figure 6: Different models generating 10,000 samples from a 2D grid of Gaussian components."No FT" stands for the discriminator not being trained to convergence.

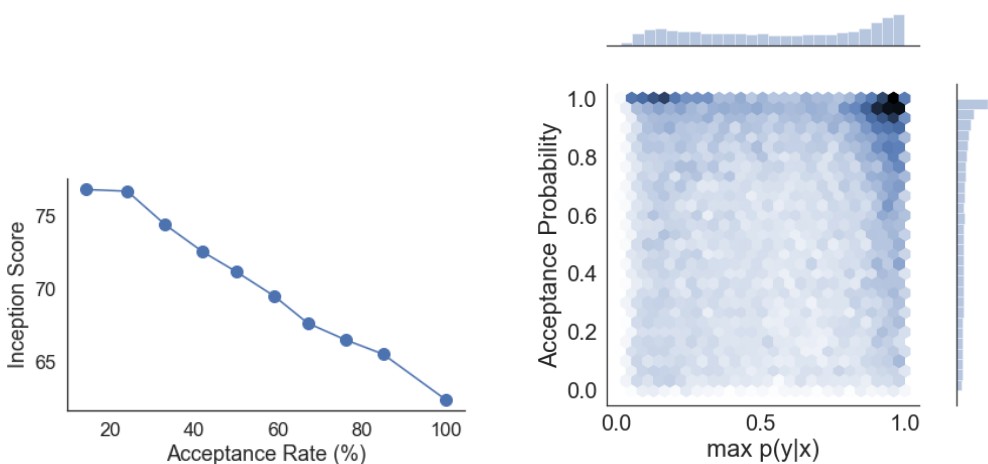

Figure 7: Inception Score versus the rate of accepting samples on average (*left*), and the acceptance probability assigned to each sample $x_i$ by DRS versus the maximum probability of belonging to one of the 1K categories based on a pre-trained Inception network, $\max_j p(y_j|x_i)$ (*right*).

## B NEAREST NEIGHBORS FROM IMAGENET

To confirm that our Discriminator Rejection Sampling is not duplicating the training samples, we show the nearest neighbor of a few visually-realistic generated samples in the ImageNet training data in Figures 8-15. The nearest neighbors are found based on their fc7 features from the pre-trained VGG16 model.

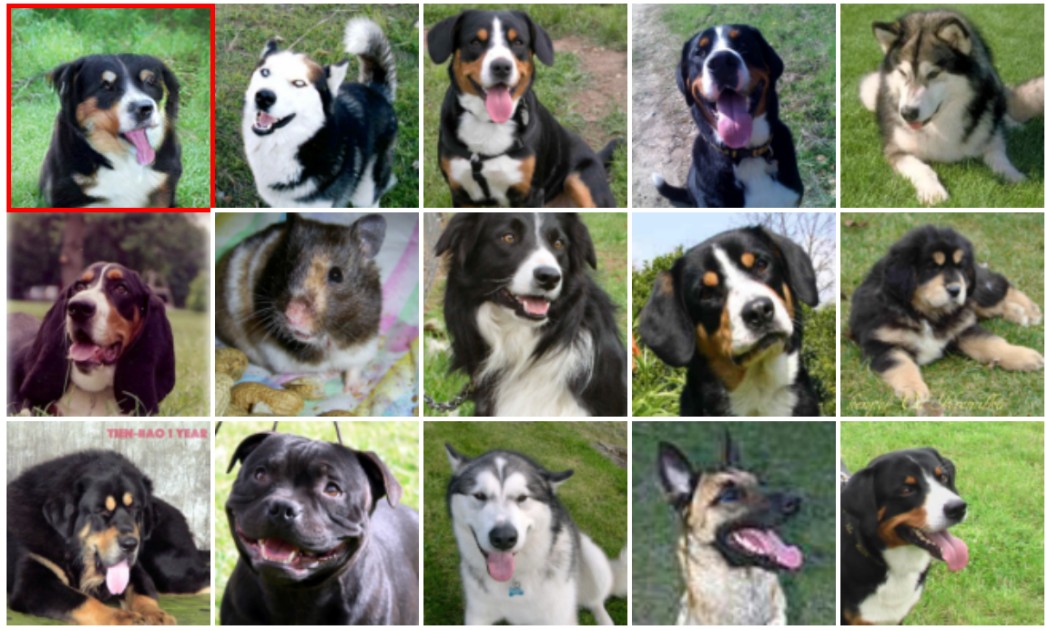

Figure 8: Nearest neighbors of the top left generated image in ImageNet training set in terms of VGG16 fc7 features

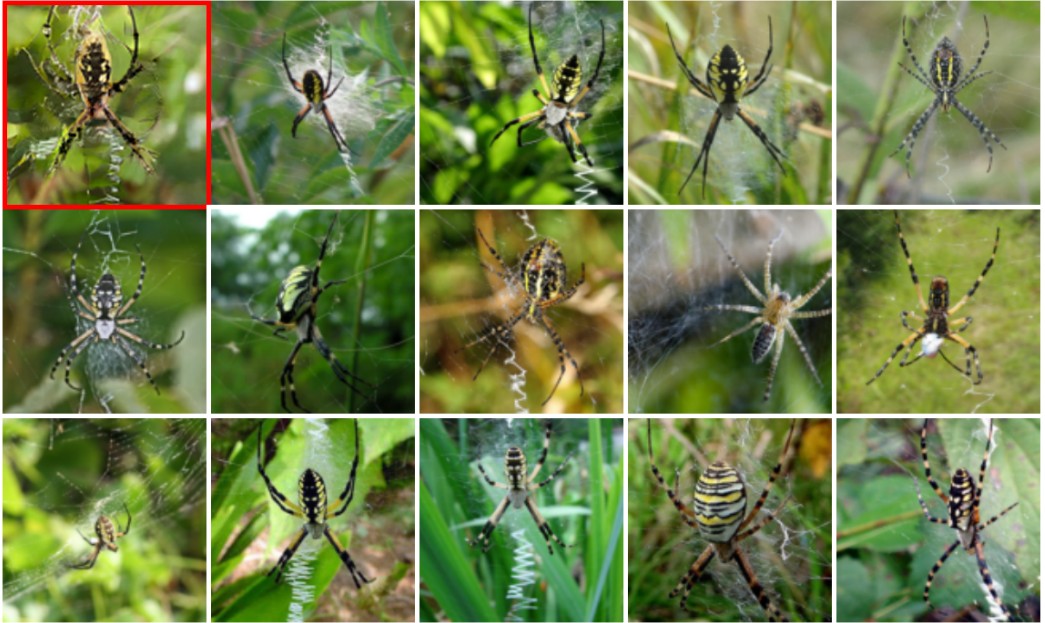

Figure 9: Nearest neighbors of the top left generated image in ImageNet training set in terms of VGG16 fc7 features

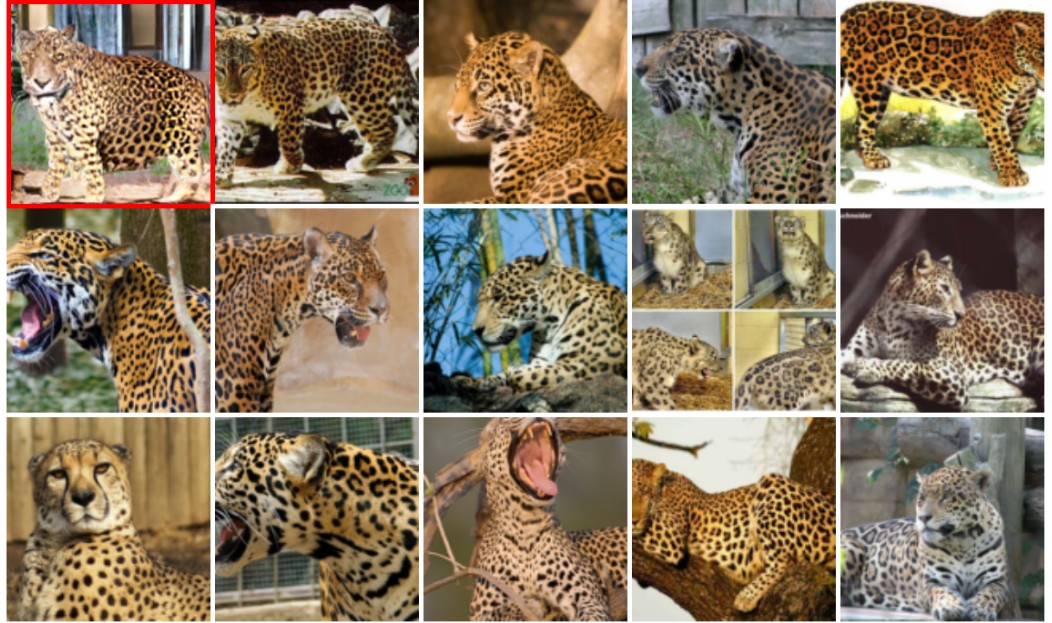

Figure 10: Nearest neighbors of the top left generated image in ImageNet training set in terms of VGG16 fc7 features

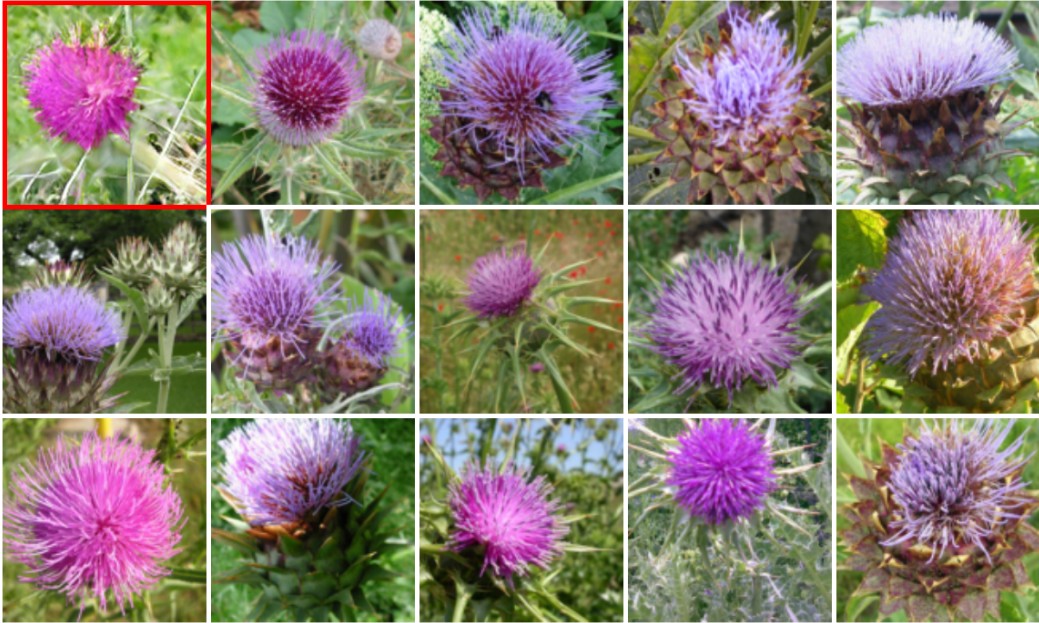

Figure 11: Nearest neighbors of the top left generated image in ImageNet training set in terms of VGG16 fc7 features

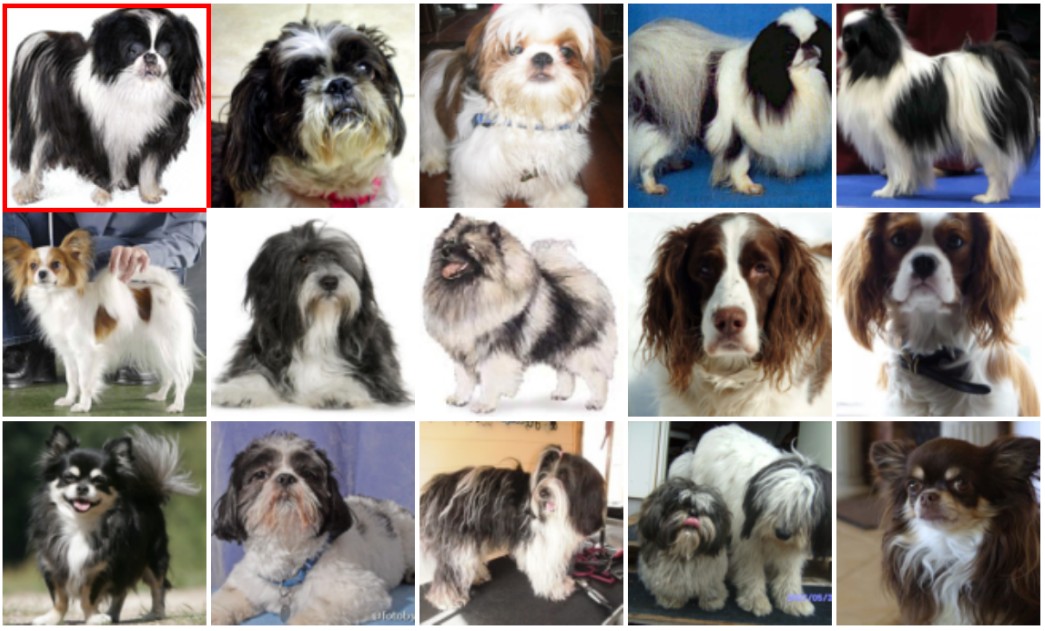

Figure 12: Nearest neighbors of the top left generated image in ImageNet training set in terms of VGG16 fc7 features

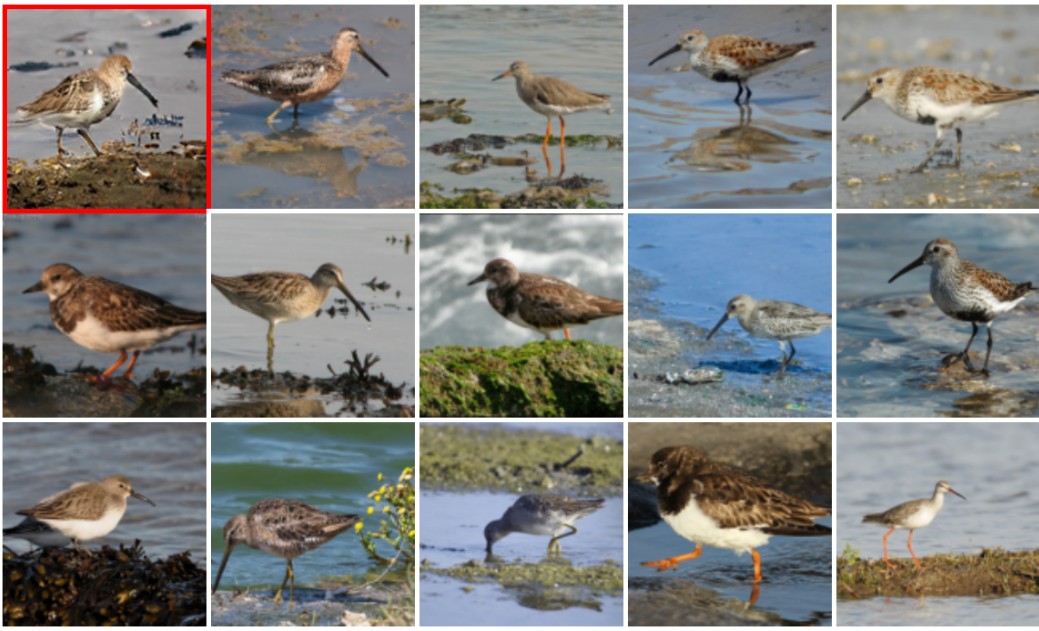

Figure 13: Nearest neighbors of the top left generated image in ImageNet training set in terms of VGG16 fc7 features

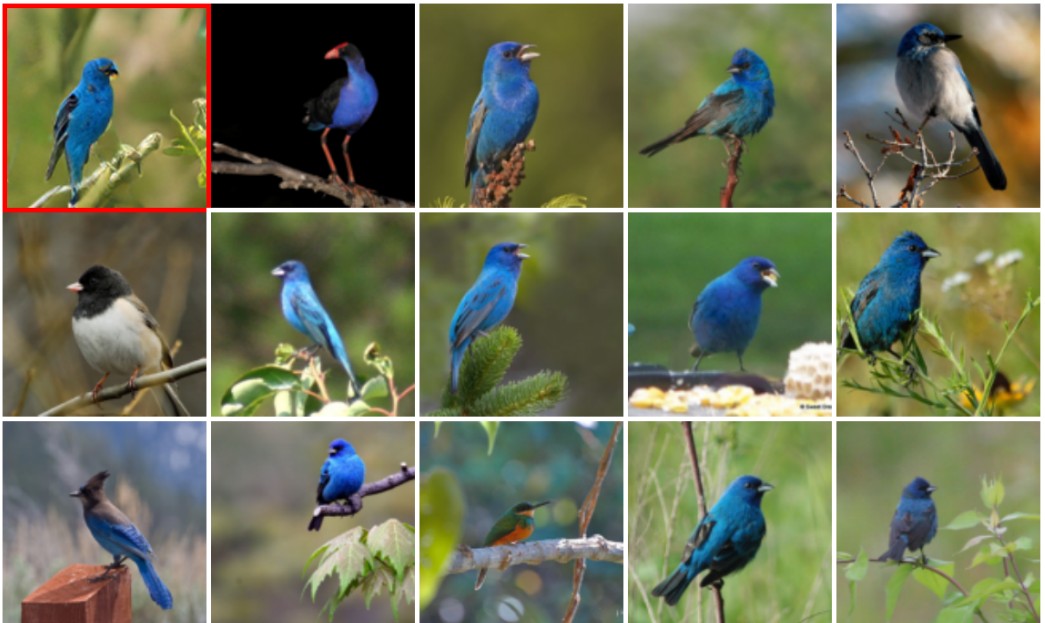

Figure 14: Nearest neighbors of the top left generated image in ImageNet training set in terms of VGG16 fc7 features

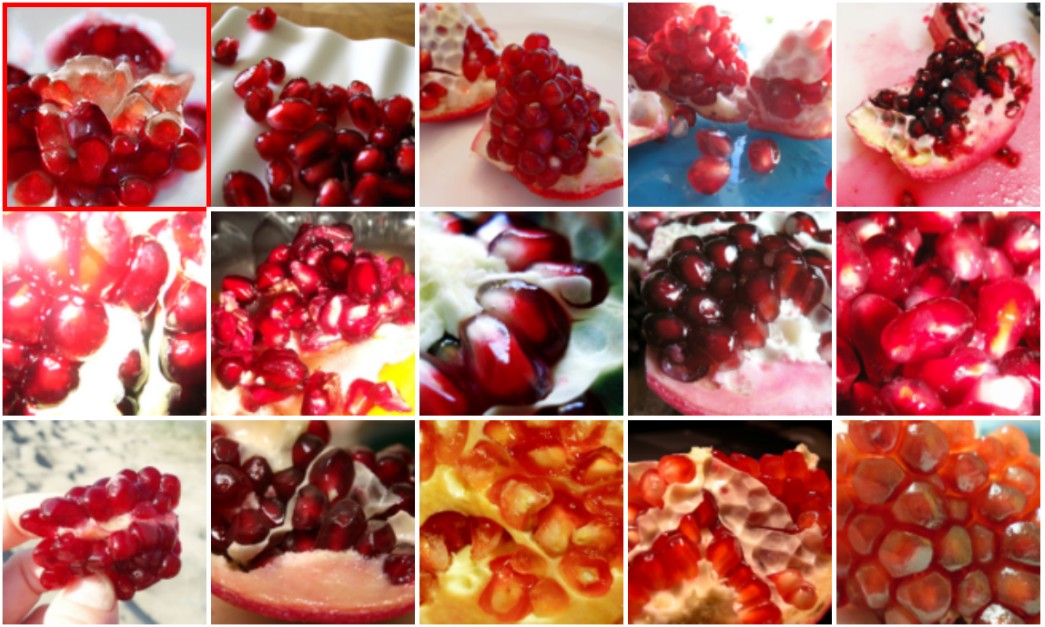

Figure 15: Nearest neighbors of the top left generated image in ImageNet training set in terms of VGG16 fc7 features

