# OpenReview forum: "Discriminator Rejection Sampling"
_ICLR.cc/2019/Conference_

### Official Review · AnonReviewer3 · 2018-11-01
**A post-processing method to filter ‘good’ generated samples for GANs**

**Rating:** 6
**Confidence:** 4

**Review:**

This paper proposed a post-processing rejection sampling scheme for GANs, named Discriminator Rejection Sampling (DRS), to help filter ‘good’ samples from GANs’ generator. More specifically, after training GANs’ generator and discriminator are fixed; GANs’ discriminator is further exploited to design a rejection sampler, which is used to reject the ‘bad’ samples generated from the fixed generator; accordingly, the accepted generated samples have good quality (better IS and FID results). Experiments of SAGAN model on GMM toys and ImageNet dataset show that DRS helps further increases the IS and reduces the FID.

The paper is easy to follow, and the experimental results are convincing. However, I am curious about the follow questions.

(1)	Besides helping generate better samples, could you list several other applications where the proposed technique is useful?

(2)	In the last paragraph of Page 4, I don’t think the presented Discriminator Rejection Sampling “addresses” the issues in Sec 3.2, especially the first paragraph of Page 5.

(3)	The hyperparameter gamma in Eq. (8) is of vital importance for the proposed DRS. Actually, it is believed the key to determining whether DRS works or not. Detailed analysis/experiments about hyperparameter gamma are considered missing.

---

> ### Author Response · Authors · 2018-11-07
> **Thanks for the review!**
>
> We thank the reviewer for his/her time and feedback. We appreciate the kind words relating to the clarity and comprehensiveness of our submission, and hope to address any remaining concerns the reviewer has here.
>
> OTHER APPLICATIONS:
>  (a) Suppose we’re designing molecules for drug discovery purposes using a generative model.
> At some point, we will have to physically test the molecules that we have designed, which could be costly.
>  If the discriminator can throw out some obviously unrealistic molecule designs, this will save us money and time.
> (b) For text generation applications, a nonsensical generated sentence in a dialog system could be rejected by the discriminator, reducing the frequency of embarrassing mistakes.
> (c) In RL applications, if we are predicting future states with a generative model, we could use this technique to throw out silly predictions, reducing the risk of taking a silly action predicated on those predictions.
> (d) More generally, you could use DRS on models that are not GANs.
>
> ADDRESSING D* ISSUE
>
> You’re right about this - we will change the wording. We don’t do anything to *fix* the problem that we can’t actually compute D*, we just show that you don’t need to precisely recover D* to get good results. The first paragraph on page 5 speculates on why this might be so, and figures 4 and 5 provide evidence for this speculation.
>
> REGARDING GAMMA:
>
> We agree that gamma is an important hyperparameter, because it modulates the acceptance rate.
> We have already made the figure you propose and have updated the PDF to include it. It is now figure 6.
> Please let us know if there are other experiments that you think would
> improve the quality of the work.

---

> > ### Comment · AnonReviewer3 · 2018-11-26
> > **Comments**
> >
> > Thanks for the interesting applications, which addressed my main concern.
> >
> > Also, I recently found that the literature [1] has also mentioned a similar resampling idea. So a relative discussion should be added into the manuscript to make clear the difference.
> >
> > [1] C. Tao, L. Chen, R. Henao, J. Feng, and L. Carin. Chi-square generative adversarial network. In ICML, 2018.

---

> > > ### Author Response · Authors · 2018-12-01
> > > **Thanks - here's our take on [1]**
> > >
> > > Thanks for bringing [1] to our attention; we hadn't seen it.
> > > We'll first summarize our understanding of the algorithm from [1] (which we'll call IR for 'Importance Resampling')
> > > and then we'll discuss differences.
> > >
> > > IR somehow computes importance weights for a set of samples using the Discriminator/Critic from a trained GAN.
> > > A single sample is drawn as follows:
> > > N samples from the trained GAN are prepared and importance weights are computed.
> > > A single one of the samples is then 'accepted' using a categorical distribution over N categories parameterized by the importance weights.
> > >
> > > The differences (between IR and DRS and between our scientific evaluation and theirs) are:
> > >
> > > 1. [1] don't continue to train D to approximate D^*.
> > > We theoretically motivate the importance of this, and we also show (in the new experiments we ran for the rebuttal) that this is important empirically.
> > > This difference may explain the small improvement given by IR (see below).
> > >
> > > 2. [1] sample one image at a time given a set of N candidates instead of the probabilistic sampling as in DRS.
> > > That is, their acceptance ratio is controlled by N.
> > > I don't think that this procedure will recover p_data given finite N?
> > > It's hard to say for sure without knowing more detail about how they are getting the importance weights.
> > >
> > > 3. We add the 'gamma trick', which you already noted is crucial to making the algorithm work in practice.
> > > Imagine that the weights of n-1 samples are tiny (e.g. e-10) and the weight of one sample is close to 1.
> > > Normalizing all of the samples by \sum{w_i} does not make any difference in the weights and thus, this importance re-sampling would not do much.
> > > The 'gamma trick' changes the acceptance probabilities such that they cover the whole range of 0 to 1 scores.
> > > This effect was also illustrated in Figure 2-A.
> > > This results in a more efficient sampling scheme when acceptance probabilities for most of the samples are very small,
> > > which happened in our ImageNet experiment (purple histogram of Figure 2-A).
> > >
> > > 4. [1] don't really provide evidence that IR yields quantitative improvement.
> > > In the supplementary material, they show a single run on which the Inception score is changed from 7.28 to 7.42, an improvement of less than 2%.
> > > Our work shows that DRS yields improvements of (61.44 / 52.34 ~ 17%) and (76.08 / 62.36 ~ 22%) respectively on the baseline and improved
> > > versions of SAGAN[2] we used for experiments.
> > > Apart from [3] (a concurrent submission to ICLR), these results are the best achieved in the literature.
> > > We think it's reasonably to expect that DRS could improve the results from [3] as well.
> > >
> > > 5. [1] seem to compare IR to a weak baseline in the experiment from the Supplementary Material.
> > > This experiment is (presumably) conducted on the unsupervised CIFAR-10 task.
> > > 7.42 is not only far from the state of the art at the time [1] was written (this is important because it gives evidence about whether IR can be 'stacked'
> > > with other improvements), but it's less than the reported performance of the main method from [1], which is given as 7.47 +/- 0.10.
> > > This is strange, because it suggests that the baseline for this experiment was not trained as well as the model in the main text (its performance of 7.28 is nearly
> > > 2 standard deviations worse).
> > > Footnote 1 in the main text says 'We used a less well-trained model and picked our samples based on the importance weights to highlight the difference.',
> > > but it's unclear if this was also intentionally done in the supplementary material.
> > >
> > > 6. [1] don't compute the FID of the accepted samples, so there is no way to know if diversity has been sacrificed for sample quality.
> > > We compute the FID and show that it has improved after DRS.
> > >
> > > 7. [1] don't provide any theoretical analysis of IR.
> > >
> > > 8. [1] don't include any illustrative toy experiments that suggest why resampling might work.
> > > We propose and give support (using the mixture of gaussians experiment) for the hypothesis that it's easier for the
> > > discriminator to tell that certain regions of X are 'bad' than it is for the Generator to avoid spitting out samples in that region.
> > >
> > >
> > > PS:
> > > We don't mean to be overly negative about [1].
> > > We understand that IR was not the primary contribution of that work.
> > > We just wish to emphasize the scope of the difference between the fraction of that work focusing on IR and our work.
> > >
> > > PPS:
> > > We saw this message after the deadline to modify the PDF.
> > > We will of course add this discussion to the final copy of the PDF when the time comes.
> > >
> > > [1] Chi-square generative adversarial network. In ICML, 2018.
> > > [2] Self-Attention GAN
> > > [3] Large Scale GAN Training for High Fidelity Natural Image Synthesis

---

### Official Review · AnonReviewer2 · 2018-11-07
**Very well written paper, with excellent results, but experiments may be unfair, and a much simpler rejection scheme may work equally well.**

**Rating:** 6
**Confidence:** 3

**Review:**

his paper assumes that, in a GAN, the generator is not perfect and some information is left in the discriminator, so that it can be used to 'reject' some of the 'fake' examples produced by the generator.

The introduction, problem statement and justification for rejection sampling are excellent, with a level of clarity that makes it understandable by non expert readers, and a wittiness that makes the paper fun to read. I assume this work is novel: the reviewer is more an expert in rejection than in GANs, and is aware how few publications rely on rejection.

However, the authors fail to compare their algorithm to a much simpler rejection scheme, and a revised version should discuss this issue.
Let's jump to equation (8): compared to a simple use of the dicriminator for rejection, it adds the term under the log.
The basic rejection equation would read F(x) = D*(x) - gamma and one would adjust the threshold gamma to obtain the desired operating point. I am wondering why no comparison is provided with basic rejection?

Let me try to understand the Gaussian mixture experiment, as the description is ambiguous:
- GAN setting: 10K examples are generated and reported in figure 3?
- DRS setting: 10K examples are generated, and submitted to algorithm in figure 1. For each batch, a line search sets gamma so that 95% of the examples are accepted. Thus only 9.5K are reported in figure 3.
- What about basic rejection using F(x) = D*(x) - gamma: how does it compare to DRS at the same 95% accept?

If this is my understanding, then the comparison in Figure 3 in unfair, as DRS is allowed to pick and choose.
For completeness, basic rejection should also be added.

Going back to Eq.(8), one realizes that the difference between DRS rejection and basic rejection may be negligible.
First order Taylor expansion of log(1-x) that would apply to the case where the rejection probability is small yields:
F(x) = (D*(x) - D*_M) + exp(D*(x) - D*_M)

x+ exp(x) is monotonous, so thresholding over it is the same as thresholding over x: back to basic rejection!

---

> ### Author Response · Authors · 2018-11-07
> **Thanks for the review!**
>
> Thanks very much for the review.
> We think that there have been two misunderstandings here, one about the Gaussian Mixture experiment and one about the purpose of the quantity F_hat(x).
> These are our fault; we should have made the paper more clear and we are modifying the draft to do so.
> In the meantime, we will address both issues here. We use > for quotes.
>
> GAUSSIAN MIXTURE EXPERIMENT:
> > - GAN setting: 10K examples are generated and reported in figure 3?
> This much is true.
>
>
> > - DRS setting: 10K examples are generated, and submitted to algorithm in figure 1. For each batch, a line search sets gamma so that 95% of the examples are accepted. Thus only 9.5K are reported in figure 3.
> This part is not true.
> You probably got confused by the line 'We generate 10,000 samples from the generator with and without DRS.' which we agree is unclear.
>
> First, we generate as many samples as needed to yield 10K acceptances, so both plots have 10k dots on them.
>
> Second, there is no line search.
> Each example is given an acceptance probability p that is generated from substituting F_hat from equation 8 for F in equation 6.
> Then, a pseudo-random number in [0,1] is compared with p to determine acceptance.
> Thus, for any given batch, the number of examples accepted is non-deterministic.
> We think that this point also relates to the misunderstanding regarding the purpose of F_hat.
>
> Third, gamma is subtracted from F.
> So setting gamma equal to the 95th %-ile value of F means that an example where F(x) is at the 95th %-ile will have a 50% chance of being accepted, because
> 1 / (1 + e^(-F_hat(x))) = 1 / (1 + e^0) = 1 / 2 in this case.
> The result is that around 23% of samples drawn from the generator made it into the final DRS plot, which means we had to draw a little less than 50k samples from the generator.
>
>
> > If this is my understanding, then the comparison in Figure 3 in unfair, as DRS is allowed to pick and choose.
> We're unsure what you mean here.
> It's true in some sense that DRS is allowed to pick and choose, but from our perspective this is part of the definition of rejection sampling?
> The generator can't figure out how to stop yielding bad samples, but the discriminator can tell which samples are bad, so we can
> throw those out and get a distribution closer to the ground truth distribution at the cost of having to generate extra samples from the generator.
>
>
> PURPOSE OF F_HAT:
> > Let's jump to equation (8): compared to a simple use of the discriminator for rejection, it adds the term under the log
> We don't think this is correct - the log already exists and we just add the gamma and epsilon terms.
> The discussion after eq 5 shows that the acceptance probability p(x) is exp(D_tilde^*(x) - D_tilde^*(x^*)).
> The tildes are important, because they mean that we are operating not on the sigmoid output of D but on the logit that is passed to the sigmoid output.
> Then we ask what F(x) would have to be s.t. 1 / (1 + e^(-F(x))) = p(x).
> This results in equation 7, *which already has the log term*.
> The only difference between F_hat and F is that we introduce the epsilon for numerical stability and the gamma to modulate the acceptance probability.
>
> > First order Taylor expansion of...
> What you say here is true, but we are not thresholding.
> We think this is the root of the misunderstanding.
> We don't consider the hard thresholding algorithm here because it might deterministically reject certain samples for which D^* is low,
> which means that we would never be able to actually draw samples from p_d, even in the idealized setting of section 3.1
>
> Please let us know if this response answers all of your questions.
> We are happy to expand.

---

> ### Author Response · Authors · 2018-12-01
> **Re: simpler rejection scheme**
>
> Please see this comment: https://openreview.net/forum?id=S1GkToR5tm&noteId=SyxH1nd7R7 or the updated PDF for experimental results on (what we think is) the simpler rejection scheme you mention.
>
> Please also let us know if there's anything else you think we can do to improve the paper quality.

---

### Official Review · AnonReviewer1 · 2018-11-09
**Good paper!**

**Rating:** 7
**Confidence:** 4

**Review:**

This paper proposes a rejection sampling algorithm for sampling from the GAN generator. Authors establish a very clear connection between the optimal GAN discriminator and the rejection sampling acceptance probability. Then they explain very clearly that in practice the connection is not exact, and propose a practical algorithm.

Experimental results suggest that the proposed algorithm helps the increase the accuracy of the generator, measured in terms of inception score and Frechet inception distance.

It would be interesting though to see if the proposed algorithm buys anything over a trivial rejection scheme such as looking at the discriminator values and rejecting the samples if they fall below a certain threshold. This being said, I do understand that the proposed practical acceptance ratio in equation (8) is 'close' to the theoretically justified acceptance ratio. Since in practice the learnt discriminator is not exactly the ideal discriminator D*(x), I think it is super okay to add a constant and optimize it on a validation set. (Equation (7) is off anyways since in practice the things (e.g. the discriminator) are not ideal). But again, I do think it would make the paper much stronger to compare equation (8) with some other heuristic based rejection schemes.

---

> ### Author Response · Authors · 2018-12-01
> **Re: comparisons w/ heuristic rejection schemes**
>
> Thanks very much for the review, please see this comment: https://openreview.net/forum?id=S1GkToR5tm&noteId=SyxH1nd7R7 for some ablation experiments and comparisons with heuristic rejection schemes.
>
> Let us know if there's anything else you think we can do to improve the work.

---

### Author Response · Authors · 2018-11-07
**FYI**

We have written individual replies to Reviews 2 and 3 (these are the only reviews at present).

We have also update the PDF to include a new figure (fig 6) on the effect of gamma.

We are working on making more updates to the draft for purposes of clarity.

---

### Author Response · Authors · 2018-11-22
**We have run the requested comparisons**

Reviewers 1 and 2 both mentioned that they would like to see comparisons to certain baselines.
We have now performed such comparisons.
We are working on adding them to the PDF, but I will discuss the results here in the meantime.

We evaluated 4 different rejection sampling schemes on the mixture-of-Gaussians dataset:

(1) Always reject samples falling below a hard threshold and DO NOT train the Discriminator to 'convergence'.

(2) Always reject samples falling below a hard threshold and train the Discriminator to convergence.

(3) Use probabilistic sampling as in eq 8 and DO NOT train the Discriminator to convergence.

(4) Our original DRS algorithm, in which we use probabilistic sampling and train the Discriminator to convergence.

In (1) and (2), we were careful to set the hard threshold so that the actual acceptance rate was the same as in (3) and (4).

Broadly speaking:
4 performs best
3 performs OK but yields less 'good samples' than 4.
2 yields the same number of 'good samples' as 3, but completely fails to sample from 5 of the 25 modes.
1 actually yields the most 'good samples' for the modes it hits, but it only hits 4 modes!

These results show that
a) continuing to train D so that it can approximate D^* (which we have already motivated theoretically) is helpful in practice.
b) performing sampling as in eq 8 (which we also motivated theoretically) is helpful in practice.

Below we provide, for each method, the number of samples within 1, 2, 3 and 4 std deviations and the number of modes hit.
For reference, we also compute these statistics for the ground truth distribution and the unfiltered samples from the GAN.

We would have liked to perform the same analysis on SAGAN, but we currently don't have access to resources that would
allow us to do this before the response deadline.

DRS ABLATION STUDY
GROUND TRUTH
Centroid coverage: 25
within 1 std: 0.3934
within 2 std: 0.8661
within 3 std: 0.9891
within 4 std: 0.9999
VANILLA GAN
Centroid coverage: 25
within 1 std: 0.273
within 2 std: 0.5308
within 3 std: 0.6615
within 4 std: 0.7561
(1) THRESHOLD NO FT
Centroid coverage: 4
within 1 std: 0.3849
within 2 std: 0.9255
within 3 std: 0.9944
within 4 std: 0.9982
THRESHOLD
(2) Centroid coverage: 20
within 1 std: 0.3478
within 2 std: 0.7023
within 3 std: 0.8359
within 4 std: 0.8928
(3) DRS NO FT
Centroid coverage: 25
within 1 std: 0.314962934062
within 2 std: 0.601736246586
within 3 std: 0.73585641826
within 4 std: 0.811841591885
(4) DRS
Centroid coverage: 25
within 1 std: 0.35277582572
within 2 std: 0.657589599438
within 3 std: 0.817463106114
within 4 std: 0.897487702038

---

> ### Author Response · Authors · 2018-11-22
> **The paper has been updated to reference these results**
>
> We have also added plots corresponding to the above values

---

### Public Comment · (anonymous) · 2019-02-03
**Nice and effective trick!**

Enjoyed reading this paper. Even implemented it for my own GAN use case (on a different dataset than the ones used in this paper) and confirm it works well!

P.S. There is an earlier related work on Variational Rejection Sampling [1], which uses rejection sampling for improving samples from the variational posterior in variational autoencoder models, using ideas similar to this paper. The difference with this work is that rejection sampling is performed in the latent space, whereas this paper focusses on the observed space. So depending on your application, it might be beneficial to use either approach!

[1] Variational Rejection Sampling
Aditya Grover, Ramki Gummadi, Miguel Lazaro-Gredilla, Dale Schuurmans, Stefano Ermon
AISTATS 2018
https://arxiv.org/abs/1804.01712

---

### Meta-Review · Area_Chair1 · 2018-12-08
**Improving GANs by rejection sampling**

**Confidence:** 4
**Recommendation:** Accept (Poster)

**Metareview:**

The paper proposes a discriminator dependent rejection sampling scheme for improving the quality of samples from a trained GAN. The paper is clearly written, presents an interesting idea and the authors extended and improved the experimental analyses as suggested by the reviewers.